# Systematic Identification of circRNAs in Alzheimer’s Disease

**DOI:** 10.3390/genes12081258

**Published:** 2021-08-18

**Authors:** Kyle R. Cochran, Kirtana Veeraraghavan, Gautam Kundu, Krystyna Mazan-Mamczarz, Christopher Coletta, Madhav Thambisetty, Myriam Gorospe, Supriyo De

**Affiliations:** 1Laboratory of Genetics and Genomics, National Institute on Aging (NIA) Intramural Research Program (IRP), National Institutes of Health (NIH), Baltimore, MD 21224, USA; kyle.cochran@nih.gov (K.R.C.); kirtanav98@gmail.com (K.V.); gkundu84@gmail.com (G.K.); krystyna.mazan-mamczarz@nih.gov (K.M.-M.); chris.coletta@nih.gov (C.C.); myriam-gorospe@nih.gov (M.G.); 2Laboratory of Behavioral Neuroscience, National Institute on Aging (NIA) Intramural Research Program (IRP), National Institutes of Health (NIH), Baltimore, MD 21224, USA; thambisettym@mail.nih.gov

**Keywords:** circular RNAs, RNA-sequencing analysis, backsplice junction

## Abstract

**** Mammalian circRNAs are covalently closed circular RNAs often generated through backsplicing of precursor linear RNAs. Although their functions are largely unknown, they have been found to influence gene expression at different levels and in a wide range of biological processes. Here, we investigated if some circRNAs may be differentially abundant in Alzheimer’s Disease (AD). We identified and analyzed publicly available RNA-sequencing data from the frontal lobe, temporal cortex, hippocampus, and plasma samples reported from persons with AD and persons who were cognitively normal, focusing on circRNAs shared across these datasets. We identified an overlap of significantly changed circRNAs among AD individuals in the various brain datasets, including circRNAs originating from genes strongly linked to AD pathology such as *DOCK1*, *NTRK2*, *APC* (implicated in synaptic plasticity and neuronal survival) and *DGL1*/*SAP97*, *TRAPPC9*, and *KIF1B* (implicated in vesicular traffic). We further predicted the presence of circRNA isoforms in AD using specialized statistical analysis packages to create approximations of entire circRNAs. We propose that the catalog of differentially abundant circRNAs can guide future investigation on the expression and splicing of the host transcripts, as well as the possible roles of these circRNAs in AD pathogenesis.

## 1. Introduction

Mammalian circular (circ)RNAs arise from the looping of the 5′ and 3′ ends of an RNA to form a covalent bond. They generally arise from backsplicing of exons [1,2], although intronic circRNAs have also been described [3]. CircRNAs can originate from both precursors of coding RNAs (mRNAs) and noncoding RNAs, and can encompass whole or parts of single exons, multiple exons, introns, and combinations of introns and exons [1,2,3,4,5,6]. With widespread use of high-throughput sequencing technologies, tens of thousands of circRNAs have been identified, typically based on the detection of their unique junction sequences, the RNA sequence where the 5′ and 3′ ends are covalently ligated [4,5]. Given their closed-loop structure, circRNAs are believed to be more stable than linear RNAs [7]. 

The functions of the vast family of circRNAs are largely unknown, but they are believed to be linked to the molecules with which circRNAs interact [1,2,4,5,6,8]. Some of the factors binding to circRNAs are microRNAs, and some abundant circRNAs may function as microRNA ‘sponges’. As a prominent example of this function, the circRNA *ciRS-7* (also known as *CDR1-AS*) is highly abundant in a range of tissues, including neurons and other brain cells, is capable of binding microRNA miR-7 at dozens of sites across the body of the circRNA, and can ‘sequester’ miR-7 [9,10]. Accordingly, decreasing the levels of *ciRS-7* caused an increase in miR-7 available in the cell for repression of miR-7-specific mRNA targets; one of these targets encodes the protein UBE2A, which is responsible for the removal of amyloid peptides present in AD brains [11]. It is important to note that the majority of circRNAs are low-abundance molecules and are unlikely function as microRNA sponges [12,13].

The interaction of circRNAs with proteins, as shown for some transcription factors and RNA-binding proteins (RBPs), may lead to changes in transcriptional and splicing programs, influence protein function, and alter the translation and stability of select mRNAs [14,15,16]. For instance, the circRNA *circMbl* arises from an exon of the *MBL/MBNL1* pre-mRNA 2; through binding to the splicing factor MBL, *circMbl* influences splicing [17]. In other examples, *circFoxo3* associated with CDK2 and p21/CDKN1A, preventing CDK2 from becoming active and halting cell cycle progression [18], while binding of *circPABPN1* to the RNA-binding protein HuR reduced the binding of HuR to cognate *PABPN1* mRNA and lowered the production of the translational activator PABPN1 [15]. In the case of the myogenesis-associated circRNA *circSamd4*, binding to PURA and PURB, repressors of myosin heavy chain (MYH) transcription, led to the derepression of MYH transcription and enabled late stages of myogenesis [16]. Finally, a few circRNAs bearing internal ribosome entry sites (IRES) may recruit ribosomes and give rise to the production of circRNA-encoded peptides, although the extent to which circRNAs are translated remains unclear [19,20,21,22]. 

Although circRNAs are found almost anywhere in the body and some are quite abundant, most circRNAs are expressed in low levels and are found only in specific tissues. Thus, many circRNAs have gained attention as reliable biomarkers of organ function, dysfunction, and disease [23,24,25,26,27,28]. Given their intrinsic stability, even if circRNAs originate in other tissues, they often eventually reach the circulation and might be found in plasma. In this study, we sought to identify circRNAs differentially abundant in Alzheimer’s Disease. We investigated available RNA-sequencing (RNA-seq) datasets from the frontal lobe, temporal cortex, hippocampus, and plasma from AD patients and healthy, age-matched control individuals that had been used previously in linear RNA analyses [29,30,31,32,33,34]. Using the circRNA identification software package CIRCexplorer2, we identified circRNAs that were differentially abundant in AD relative to normal controls across the datasets. Their predicted sequences were reconstructed with the CIRCexplorer2 *de novo* assembly module in order to begin to examine possible interaction partners. Although plasma circRNAs did not appear to reflect brain circRNAs, an interesting group of brain circRNAs was found to originate from parent genes implicated in synaptic plasticity and neuronal survival (*DOCK1*, *NTRK2*, and *APC*) and others implicated in vesicular traffic (*DGL1*/*SAP97*, *TRAPPC9*, and *KIF1B*). We propose that the circRNAs identified in this study can guide the analysis of specific processes aberrant in the AD environment.

## 2. Materials and Methods

### 2.1. RNA Sequencing Datasets from Brain Tissue and Plasma

Six datasets containing total RNA-seq data from AD patients and cognitively normal, age-matched controls were selected from the Gene Expression Omnibus (GEO). Each RNA-seq dataset selected contained both AD and control human brain tissue or blood samples. Altogether, four studies with postmortem brain tissue including two from the frontal lobe (GSE53697, GSE110731), hippocampus (GSE67333), and temporal lobe (GSE104704) [29,30,31,32], and two studies from plasma (GSE161199, PRJNA574438) obtained from live subjects [33,34] were found to be suitable for circRNA-seq analysis. Each of the six studies included multiple age- and sex-matched samples, 435 in total: 213 from AD patients and 222 from matched control individuals (80 brain samples and 355 plasma samples; Figure 1A).

### 2.2. Aligning FASTQ Files to the Human Genome

SRA (sequence read archive) files from each study were downloaded from GEO and converted to FASTQ files using the SRA Toolkit [35]. These FASTQ files were then aligned to the human genome (HG19, Ensembl v82) [36] using the STAR aligner (v2.7.1a) and the TopHat2 aligner (v2.1.1) [37,38] (Figure 1B). The alignment process produced Sequence Alignment Map (SAM) and Binary Alignment Map (BAM) files, as well as Chimeric.out.junction files, which denote specific circRNA junction sequences. It was important to determine whether the sample had been subjected to paired-end or single-end sequencing, as each required different alignment parameters.

### 2.3. Using CIRCexplorer2 to Analyze Junction Reads

SAM files generated by the STAR aligner containing mapped sequencing information were processed through the CIRCexplorer2 (v2.3.8) pipeline. First, the Chimeric.out.junction files were parsed using the CIRCexplorer2 *parse* module creating BED (Browser Extensible Data) files containing backspliced junction information. The BED files were then annotated with information from the reference genome (Ensembl v82) which produced GTF (gene transfer format) files containing the location, strandedness, parent gene, and number of reads for each circRNA identified. 

### 2.4. Constructing circRNA Body Approximations with De Novo Assembly

The BAM files generated by TopHat2 were processed through the CIRCexplorer2 assemble module which used the RABT (reference annotation-based transcript) method to create assembled circRNA body approximations [39]. CIRCexplorer2 starts at the known backspliced junction sequence and builds outwards using the reference genome to create an approximation of the entire body of the circRNA. The circRNA body approximations were then annotated using the CIRCexplorer2 *de novo* module which attempts to identify novel circRNAs, circRNA isoforms, and alternative splicing events [40]. These annotated body approximations are stored in a GTF file similar to the annotated backspliced junction reads file created by the *parse* module.

### 2.5. Gene Expression Analysis in R

The GTF files containing the annotated circRNA information were analyzed in R using packages such as Rsubread (v1.22.2) and edgeR (v3.32.1) [41,42]. To filter out noise, circRNAs which showed expression of ≥0.125 counts per million (CPM) in at least two of the samples in each cohort were used for further analysis. The raw read counts of the subset were log-transformed (log_2_) and normalized by the Trimmed Mean of M-values (TMM) method. Then a negative binomial generalized linear model (GLM) was fit with gene expression as the dependent variable and clinical cohort as the predictor variable. All circRNAs that exhibited a three-fold change or greater (|logFC| ≥ 1.5) between AD and control, and statistically significant *p*-values (*p* < 0.05) were deemed to display significantly different expression levels. This analysis was also performed on the identified linear RNAs. All differentially abundant RNAs, both circular and linear, were then compiled with their fold change, *p*-value, parent gene symbol, and Ensembl transcript ID. These files were then cross-referenced with each other to identify shared differentially abundant circRNAs (Appendix A) and linear RNAs in AD relative to control across all studies.

### 2.6. Relating circRNAs to AD Pathology

CircRNAs displaying differential abundance in AD patients were then annotated with their corresponding Ensembl transcript ID (for linear RNAs) and cross-referenced among the studies to generate lists of commonly differentially abundant circRNAs across the four brain studies and two plasma studies. In multiple cases, many circRNAs were found to be associated with a single parent gene and transcript. The parent genes of the circRNAs that were differentially expressed (AD vs. normal) in at least 2 out of 4 brain tissue studies, or both plasma studies, were investigated (Figure 2). The overlap between studies was determined by assignment to the parent gene, due to the undetermined function of individual circRNAs. 

## 3. Results

### 3.1. Analysis of AD-Associated RNA-Seq Datasets to Identify circRNAs

We set out to identify circRNAs selectively associated with AD pathology. In order to analyze the maximum number of samples, we searched the literature for studies that reported RNA-seq analyses from persons with AD and matched controls, with care that the total RNA (poly-A and non-poly-A RNA) sequences were available, so that we could extract the junction sequences and identify circRNAs. We focused on human brain tissues obtained at necropsy, as AD-associated circRNAs might reveal important molecular details of AD biology and pathogenesis. As summarized in Figure 1A, four studies were found in the literature that reported brain tissue RNA-seq analyses following these criteria: two from the frontal lobe (GSE53697, GSE110731), one from the hippocampus (GSE67333), and one from the temporal lobe (GSE104704) [29,30,31,32]. In all, RNA-seq was identified and retrieved for 80 brain samples—35 from AD patients, 45 from cognitively normal individuals. We also focused on plasma, where circRNAs may reveal other important AD biology, possibly from tissues outside of the brain, and may thus uncover possible biomarkers for AD. As indicated in Figure 1A, we identified two main studies in plasma following the above criteria (GSE161199 and PRJNA574438), totaling 355 individuals—178 with AD, 177 normal controls [33,34].

Following the pipeline described in Figure 1B, we first collected relevant datasets that compared AD and control samples from human subjects, retrieved the FASTQ files from each sample, and analyzed the FASTQ files using two strategies. In one arm of the analysis, using the STAR aligner, FASTQ files were aligned it to the human genome, which yielded files ‘Chimeric.out.junction’ and ‘accepted.hits’ that contained the backspliced junction sequences of circRNAs and linear RNA sequences, respectively. Reads were aligned as presented in Appendix A. Next, from the Chimeric.out.junction files, we used CIRCexplorer2′s *parse* and *annotate* modules to create datasets containing annotated circRNAs; other aligners might reveal different circRNA datasets. Thousands of circRNAs were identified from each study (Appendix A). Similarly, we utilized the featureCounts package in R to annotate the linear RNA sequences. We then analyzed the annotated circRNA and linear RNA datasets in R, while looking for circRNAs differentially abundant and identifying their parent genes.

In the other arm, we used a different aligning software, TopHat2, but followed the same general process to create a file that contained the backspliced junction information. TopHat2 was used because it is required of the CIRCexplorer2 *assembly* and *de novo* modules. These modules took the backspliced junction datasets and constructed an approximation of the complete body of the circRNAs. Once the assembled body approximations were created, they were annotated and analyzed in R.

### 3.2. Analysis of circRNAs Differentially Abundant in AD Brain Compared with Control Brain

Focusing first on the four studies that used brain tissue (GSE53697, GSE110731, GSE67333, and GSE104704) [29,30,31,32], ~3700 circRNAs were found to display different abundance between normal and AD in at least one study, setting a threshold of at least a three-fold change in expression in either direction; about 2000 circRNAs were higher in AD relative to controls, while 1700 were lower relative to controls. For each of the four studies, volcano plots were created indicating the relative differential expression and significance for each circRNA (Figure 2). When comparing AD with control groups, the top 10 circRNAs (based on fold change) most abundant and least abundant circRNAs are listed in the table displayed in each panel, along with the gene symbol of the parent transcript and the transcript ID (Figure 2). Taken together, these results included numerous circRNAs showing differential abundance in brains of AD relative to control groups, with comparable numbers of circRNAs showing higher and lower abundance in AD relative to control.

Among the circRNAs showing increased abundance in AD relative to Control, 10 circRNAs were shared across the four studies, and 43 were found elevated in at least three studies. Other circRNAs were found elevated in AD in only two studies or one study (Figure 3A, Appendix A). Among the circRNAs showing decreased abundance in AD relative to Control, only three circRNAs were shared across the four studies (originating from loci *ANKS1B*, *ARHGAP26*, and *DPYD*), and 19 were found lower in at least three studies. The remainder of circRNAs were found reduced in AD in only two studies or one study (Figure 3B, Appendix A). These results suggest that in different brain regions of AD individuals, there may be more consistency in the circRNAs that are significantly elevated than in circRNAs that are significantly less abundant.

### 3.3. Analysis of circRNAs Differentially Abundant in AD Plasma Compared with Control Plasma

Turning our attention to the plasma studies (GSE161199 and PRJNA574438) [33,34], and employing the same threshold of greater than three-fold change in either direction, ~1700 circRNAs displayed significantly different abundance in AD relative to control individuals: ~350 circRNAs were significantly more abundant and ~1100 were significantly less abundant. Volcano plot representations of the circRNAs in each study are shown, along with the top higher and lower circRNAs in each RNA-seq collection (Figure 4A,B). In all, 15 circRNAs had significantly increased abundance in both studies (Figure 4C, *left*), but only one was found to have significantly decreased abundance in both these studies (Figure 4C, *right*). The aggregate of differentially abundant circRNAs (higher as well as lower in AD relative to normal) are displayed in Appendix A. Although the 15 elevated circRNAs originated from genes without any reported relation to AD, they could be good biomarkers for AD. We note that the intersection of the Venn diagram in Figure 4C shows a lower overall number of dysregulated circRNAs relative to the sum of intersections in Appendix A because the conditions of overlap are more stringent in Figure 4C, as we required the circRNAs to be dysregulated in the same direction, either upregulated in both studies, or downregulated in both studies. In a few cases, circRNAs were found dysregulated in two studies, but upregulated in one and down regulated in the other. 

When each plasma study was individually compared with the circRNAs that were found as commonly differentially abundant (in at least 3 of the 4 brain studies) we identified an overlap of 49 circRNAs in PRJNA574438 (Figure 5A, *right*) and 4 in GSE161199 (Figure 5A, *left*). These circRNAs are listed in Appendix A. Lastly, in Figure 5B, all six studies were compared showing differentially abundant circRNA through every possible combination. Out of the 53 circRNAs found in 3 out of 4 brain studies and at least 1 plasma study, 2 circRNAs, originating from gene loci *KIF1B* and *DLG*, were found to be related to AD. The list of all circRNAs found in different brain regions that are also found in plasma is shown (Figure 5B, Appendix A). These detailed lists can help to identify biomarkers for AD pathology from specific brain regions. As more studies in which total RNA is analyzed without poly(A) selection, we may see more balanced differences across the datasets. It will also be important, in future work, to confirm the value of circRNAs as biomarkers in these human samples using molecular biology methods that include RNase R digestion of linear RNA. 

### 3.4. Predictions of Full circRNA Body Sequences

Recent bioinformatic tools (e.g., CIRCexplorer2, CIRI, find_circ) have enabled the prediction of *de novo* body approximations for circRNAs, including the identification of circRNA isoforms. An important caveat is that without being able to subtract poly(A)-containing RNA, it is not possible to know with certainty if a given RNA-seq read was originated from RNA that was linear or circular. We compared circRNA sizes just based on sequence reads with circRNA sizes after *de novo* approximations to predicted body sizes. Interestingly, the size of the circRNAs calculated from canonical linear RNA splicing obtained from database largely matched those calculated by *de novo* assembly (Figure 6A, including all 6 studies, GSE53697, GSE110731, GSE67333, GSE104704, GSE161199, PRJNA574438), supporting the notion that the body of the circRNA is not too different from the linear RNA from which it is generated. Nonetheless, in some cases, alternative exon usage led to different circRNA isoforms, as shown for two circRNAs from chromosomes 1 and 14, differentially abundant in brain (Figure 6B), where a number of exons (teal, green, purple) were alternative included in the circRNA. These putative circRNAs need to be confirmed by other methods before they are studied further.

### 3.5. Top Differentially Abundant circRNAs Originated from Genes Implicated in AD Pathogenesis

Among the circRNAs differentially expressed in 2, 3, or all 4 of the brain studies, we identified six with known associations with AD pathology (Figure 7) [43], originating from genes *DOCK1*, *NTRK2*, *DLG1*, *TRAPPC9*, *APC*, and *KIF1B*. The circRNA coordinates, the number of studies in which they were differentially expressed, the number of circRNAs higher and lower in AD relative to control (direction of changes), and the links to AD pathobiology are listed. Dysregulated expression from the gene *DLG1*, from which one of the 6 circRNAs arises (*hsa_circ_chr3_196802707_196817897_R*) is thought to be directly associated to neuronal strength and memory loss, while other genes, such as *DOCK1* (from which circRNA *hsa_circ_chr10_128859931_128908618_F* arises) or gene *NTRK2* (from which circRNA *hsa_circ_chr9_87482157_87482346_F* arises), are connected in a broader neurological context [43]. For these 6 circRNAs, the direction of change was shared across the groups, except for the 12 circRNAs originating from *DOCK1*, eleven of which were elevated, but one was decreased. 

Additionally, a few circRNAs that were shared across samples and were known to associate to AD pathology were present in plasma. Of the circRNAs differentially expressed across at least two of the brain studies, 367 were found differentially expressed in at least one of the plasma studies when including circRNAs that exhibited a lower threshold (>2-fold) level of change. Included in these 367 were circRNAs from parent genes *DOCK1*, *DLG1, TRAPPC9, KIF1B,* and *APC*. When using a three-fold threshold, *KIF1B* and *DLG1* continued to appear in at least one plasma study and three of the four brain studies. All differentially expressed circRNAs which were common in both the plasma samples had no documented associations to AD or neurological dysfunction, although the presence of the aforementioned dysregulated parent genes could mean they might act as biomarkers for AD in blood.

### 3.6. Linear vs. circRNA Expression in AD Samples

Finally, we compared the expression of linear RNAs (mRNAs) and circRNAs originated from the same gene regions. In aggregate, there was little overlap in genes giving rise to significantly changed circRNAs and genes giving rise to significantly changed linear RNAs. Although some circRNAs correlated positively with the abundance of their linear counterpart (e.g., *hsa_circ_chr10_116879948_116931050_F*), other circRNAs correlated negatively (e.g., *hsa_circ_chr13_78293666_78327493_F*). In a specific sample from GSE53697, some abundantly expressed circRNAs were found to also have abundantly expressed linear counterparts, while other some abundantly expressed circRNAs had linear counterparts that were only minimally expressed. In Figure 8A, we compared the number of circRNA reads (in CPM) and to the number of linear RNA counts (in CPM), attempting to find a correlation between the abundance of circRNAs and linear RNAs. The lack of overall correlation suggests that, in general, there is no simple relationship between the abundance of a linear RNA and the abundance of a circRNA. Expression of only a subset of circRNAs (red box in Figure 8A) is linked to the abundance of its linear RNA counterpart whereas another subset of circRNAs (blue box in Figure 8A) appears to be present even when the linear RNA is not highly abundant, probably due to higher stability of the circRNA. In Figure 8B, *left,* we list some examples of parent genes that have circRNAs with high read counts as well as linear RNAs with high read counts; in Figure 8B, *right*, we list genes with high numbers of circRNA read counts but low abundance of linear RNA read counts.

## 4. Discussion

In this study, we have identified circRNAs in previously reported RNA-seq datasets from brain (hippocampus and temporal and frontal lobes [29,30,31,32]) and plasma [33,34] from individuals with AD and matched cognitively normal controls. While these earlier reports focused on the linear RNAs expressed in these groups, here we sought to identify junction sequences, which are specific to circRNAs and were not previously examined. Using threefold change and (*p* < 0.05) as inclusion criteria, our analysis revealed that many circRNAs were differentially abundant in AD brain relative to normal control brain (Figure 2); the direction of the change differed, as some were higher in AD, while some were higher in normal controls. In plasma, the comparison between AD and normal from both studies showed a few shared circRNAs that were more abundant in AD, while only one circRNA was lower in both plasma studies (Figure 4). However, when compared to individual plasma studies there was more significant overlap: 83 distinct circRNAs (Figure 5A, Appendix A) from distinct gene loci were found to be differentially abundant in both brain and plasma, and some of them have known associations with AD.

The reduced number of circRNAs shared across the different brain regions is in keeping with the fact that many circRNAs are produced in specific cell types, tissue regions, metabolic states, and developmental stages [7,53,54]. Thus, in the brain, a multitude of cell types (including neurons, microglia, astrocytes, oligodendrocytes, and stroma cells) may have contributed circRNAs to the collections analyzed (80 samples total), thereby widening the range of possible circRNAs present in the sample. Moreover, plasma circRNAs are potentially generated in tissues across the whole body and entered the bloodstream; such sources include circulating cells, cells lining the circulatory system, and cells deeper in organs that can pour RNA content (e.g., via extracellular vesicles or as a result of cell death [55]) into the circulation. The overlap across plasma samples in the two studies analyzed (355 donors total), would be expected to be even smaller, as was the case in our analysis (Figure 4). While the overlap between plasma and brain studies was not very robust, there were still circRNAs found to be differentially abundant in both tissue types. To search for potential biomarkers in the plasma study we lowered the log fold-change threshold from a three-fold change to a two-fold change. By doing so, five out of the six circRNAs associated with AD genes appeared in plasma (*DOCK1*, *DLG1*, *KIF1B*, *APC*, and *TRAPPC9*). Given the rise in interest in RNA biomarkers of neurodegenerative disease [56], we hoped to find some plasma circRNAs that might be diagnostic or prognostic for AD. Our analysis and that from Dube et al. [56] found some similar circRNAs associated to be differentially abundant in varying capacities including circRNAs derived from *DOCK1*.

In addition, the circRNAs differentially abundant in AD brain relative to normal brain pointed to regulatory gene expression programs potentially relevant to AD pathobiology. One set of circRNAs originated from genes that have been implicated in neuronal development, synaptic plasticity, and neuronal survival. It was interesting to see a rise in circRNAs derived from the *DOCK1* locus, which encodes a member (DOCK1, also known as DOCK180) of a family of 11 ‘dedicator of cytokinesis’ proteins implicated in activating small GTPases [44]. Although it is widely expressed, in the neural tube, DOCK1 was shown to participate in the axonal outgrowth elicited by the protein netrin and implicating the signaling protein RAC1 [45]. *DOCK1* was an AD-associated gene in a module of polygenic risk scores associated with astrocyte, language, and cognitive decline [46]. The specific function of circRNAs derived from this locus is unknown at present, but could be indicative of astrocyte dysfunction in AD brain environment. The gene *NTRK2* encodes the protein ‘neurotrophic tyrosine kinase receptor type 2′, a member of a family of NTRKs that internalize signals from brain-derived neurotrophic factor (BDNF) and affect neuronal development, plasticity, and survival. *NRTK2* has been genetically associated with AD [47,48], although no circRNAs derived from transcripts expressed from this locus have been reported. Another gene giving rise to differentially abundant circRNAs, *APC* (Anaphase Promoting Complex), is also implicated in synaptic plasticity, and neuronal development and survival [51]. 

The other set of circRNAs originated from genes encoding proteins important for intracellular transport. Discs large homolog 1 (DLG1), also known as synapse-associated protein 97 or SAP97, is involved in vesicular trafficking and has been linked to a number of brain pathologies [49]. Interestingly, TRAPPC9 (Trafficking Protein Particle Complex Subunit 9) is also implicated in vesicular transport and has similarly been linked to cognitive impairment [50]. The final gene from which differentially abundant circRNAs were generated (*KIF1B*) (Figure 7) was also implicated in axonal transport and vesicular traffic [52]. As with other genes in this short list, circRNAs derived from the *DGL1*/*SAP97*, *TRAPPC9*, and *KIF1B* loci have not been reported previously, so it is not clear at present whether altered abundance of these circRNAs reflects changes in overall transcription of the host genes or in splicing of host precursor RNAs. Moreover, whether they are passive byproducts or have active functions in the AD brain also remains to be investigated.

Some differentially abundant brain circRNAs were also differentially abundant in blood, making them potential biomarkers for AD. *KIF1B* and *DLG1* specifically are good candidates for biomarkers as they appeared in brain samples and plasma samples even using a three-fold change threshold. Investigating biomarkers for AD in plasma offers a potential path to detecting AD less invasively and more efficiently.

Further work is also needed to investigate possible functions of the circRNAs differentially abundant in AD vs. normal brain. First, it will be informative to know the specific cell types in which these circRNAs are found, possibly using modified single-cell analysis with sufficient depth to detect circRNAs and assess the number of copies that exist per cell. Second, it is important to elucidate the entire body of the circRNAs, so that confident predictions of interacting factors can be made, particularly for abundant circRNAs. Although the bioinformatic tools used in this study offered circRNA whole-body predictions, whole-body sequences should be determined rigorously using alternative methods such as Oxford Nanopore Technologies sequencing. Finally, molecular validation of factors interacting with the circRNAs differentially expressed in AD brain will shed light into some of the possible roles described for circRNAs [1,2,4,5,6,8]. It can then be assessed whether such circRNAs might associate functionally with proteins, microRNAs or other nucleic acids, perhaps function in scaffolding or sequestration of factors, or even in partial translation. These results can offer important information regarding the role of specific circRNAs in the AD environment, and point to specific future areas of therapeutic intervention in AD.

## Figures and Tables

**Figure 1 genes-12-01258-f001:**
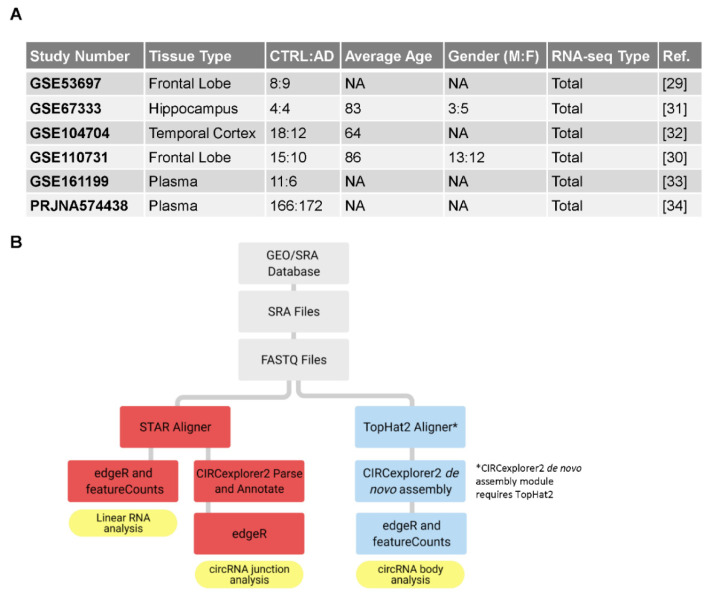
Datasets and workflow used in this study. (**A**) Each study that was used for gene expression analysis, including the brain region, number of samples, RNA sequencing type, and available subject demographics. (**B**) Workflow chart, starting with total RNA-sequencing data, alignment to the human genome, identification of circRNAs, and measurement of differential abundance.

**Figure 2 genes-12-01258-f002:**
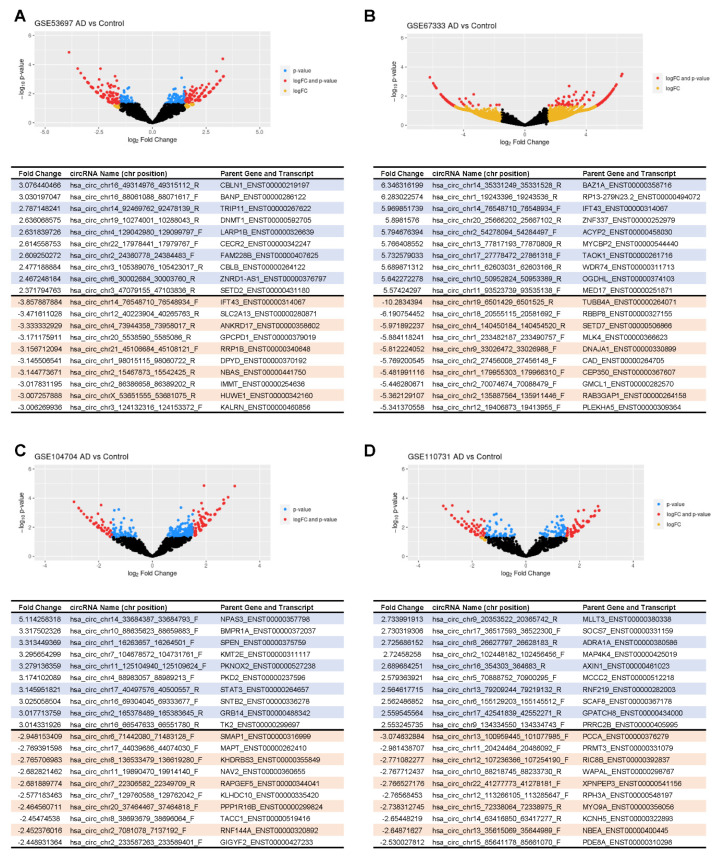
Differentially abundant circRNAs in brain datasets. Volcano plot representations of the four studies using brain tissue samples included in this analysis: frontal lobe (GSE53697) (**A**), hippocampus (GSE67333) (**B**), temporal lobe (GSE104704) (**C**), and frontal lobe (GSE110731) (**D**). Each point on these plots represents a specific circRNA with log fold-change (logFC) on the x-axis and −log_10_ (*p*-value) on the y-axis when comparing AD samples with normal control individuals. The points in yellow represent circRNAs with only logFC greater than 1.5 (threefold higher), the points in blue represent circRNAs with only *p*-value < 0.05 (i.e., −log_10_ (*p*-value) > 1.3), and the points in red represent circRNAs both logFC > 1.5 (threefold lower) and *p*-value < 0.05. Below each volcano plot is a list of the top 10 most highly abundant (blue) and top 10 least abundant (orange) when comparing circRNAs in brain from AD individuals to cognitively normal control individuals.

**Figure 3 genes-12-01258-f003:**
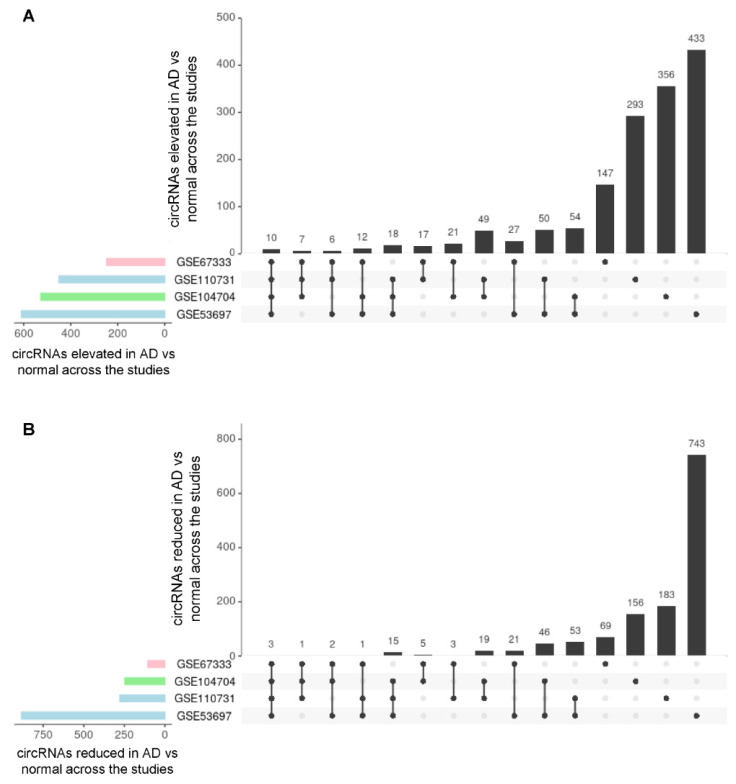
Shared circRNAs differentially expressed across studies. Each bar and corresponding filled in dots below represent the number of circRNAs more highly abundant in AD than control (**A**) and less abundant in AD than control (**B**) and the studies in which they were found (see also Appendix A). For example, in (**A**) 10 circRNAs were elevated across all four studies; in (**B**) only 3 circRNAs were less abundant in all four studies (see also Appendix A).

**Figure 4 genes-12-01258-f004:**
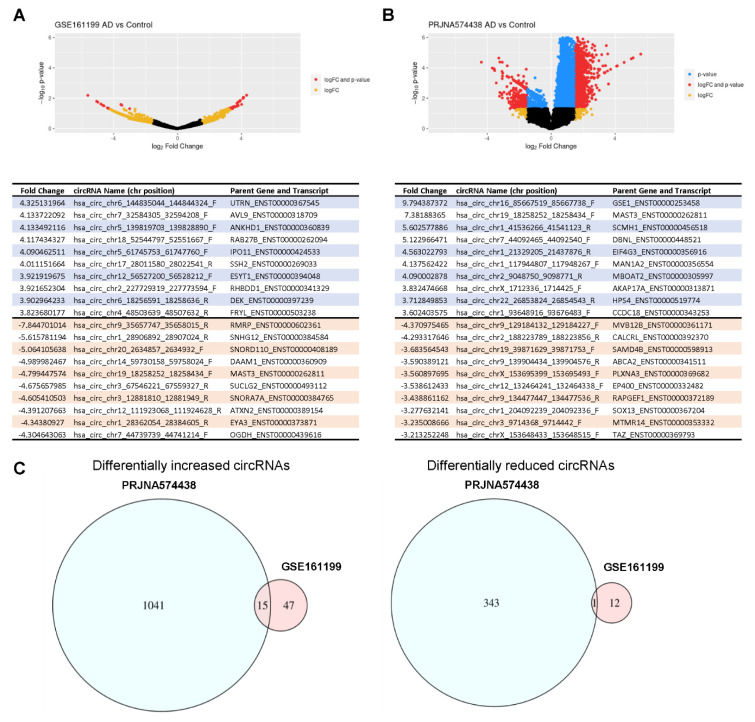
Differentially abundant circRNAs in plasma datasets. Volcano plot representations of the four studies using plasma study groups included in this analysis: GSE161199 (**A**) and PRJNA574438 (**B**). As explained in Figure 2, each point on these plots represents a specific circRNA with log fold-change (logFC) on the x-axis and *p*-value on the y-axis when comparing AD samples (plasma) with normal control individuals. The points in yellow represent circRNAs with only logFC greater than 1.5 (threefold higher), the points in blue represent circRNAs with only *p*-value < 0.05 (i.e., −log_10_ (*p*-value) > 1.3), and the points in red represent circRNAs both logFC > 1.5 (threefold lower) and *p*-value < 0.05. Below each volcano plot is a list of the top 10 most highly abundant (blue) and top 10 least abundant (orange) when comparing circRNAs in plasma from AD individuals relative to normal control individuals. (**C**) Venn diagrams represent the number of circRNAs in plasma more highly abundant in AD (*left*) and less abundant in AD (*right*) compared with control, and the studies where they were found.

**Figure 5 genes-12-01258-f005:**
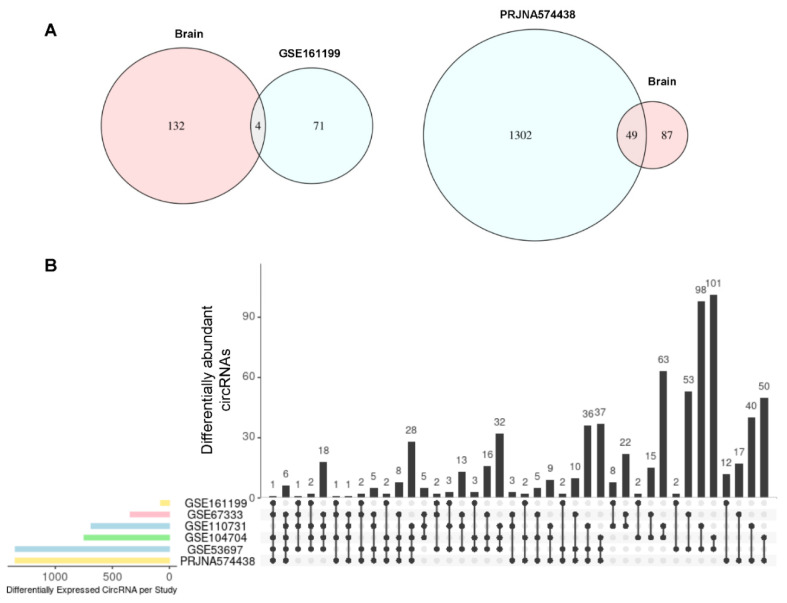
Common differentially expressed circRNAs across all studies. (**A**) Two Venn diagrams depicting differentially expressed circRNAs that were found in at least three of the four brain studies and each plasma study individually. (**B**) An upset plot (see Figure 3 legend) showing the differentially expressed circRNAs (with expression levels significantly increased or decreased) that were shared across all studies, brain and plasma.

**Figure 6 genes-12-01258-f006:**
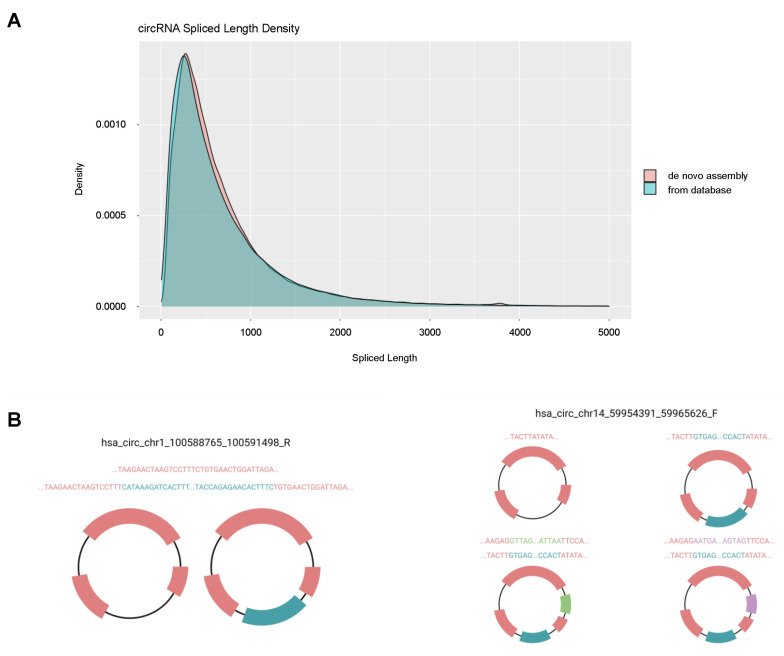
Spliced length density plot. (**A**) The spliced lengths of circRNAs calculated using the STAR aligner on FASTQ files obtained from the GEO/SRA public database is shown (aqua) compared to the density of spliced lengths of circRNAs from the *de novo* assembled body approximations (pink). The extensive overlap of the two curves suggests that the body size approximations of the circRNAs in this study are only slightly larger than those determined from gene models in the GEO/SRA database. (**B**) Two examples of circRNAs displaying alternative *de novo* assembled isoforms: hsa_circ_chr1_100588765_100591498_R and hsa_circ_chr14_59954391_59965626_F. These isoforms were created by exon skipping and not picked up by the default module of CIRCexplorer2.

**Figure 7 genes-12-01258-f007:**
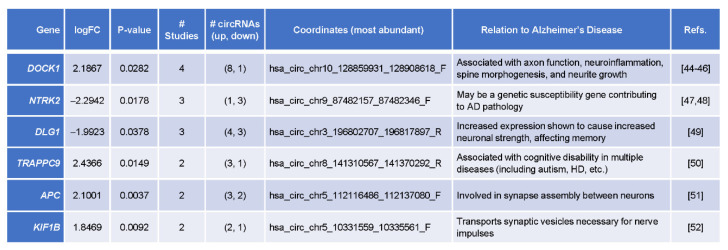
Differentially abundant circRNAs shared across studies, originated from AD-associated gene loci. Table summarizes circRNAs differentially abundant in brain samples from AD relative to normal individuals, produced from parent RNAs that were transcribed from gene loci linked to AD pathology. The gene names (‘*Gene*’), relative levels and significance for circRNAs displaying the greatest differences in abundance (‘*logFC*’, ‘*p-value*’), number of studies in which the circRNAs were found differentially abundant (‘*# Studies*’), the circRNAs more abundant (up) and less abundant (down) in AD relative to normal, the genomic location (‘*Coordinates*’), and the ‘*Relation to Alzheimer’s Disease*’, and the pertinent references are indicated [44,45,46,47,48,49,50,51,52].

**Figure 8 genes-12-01258-f008:**
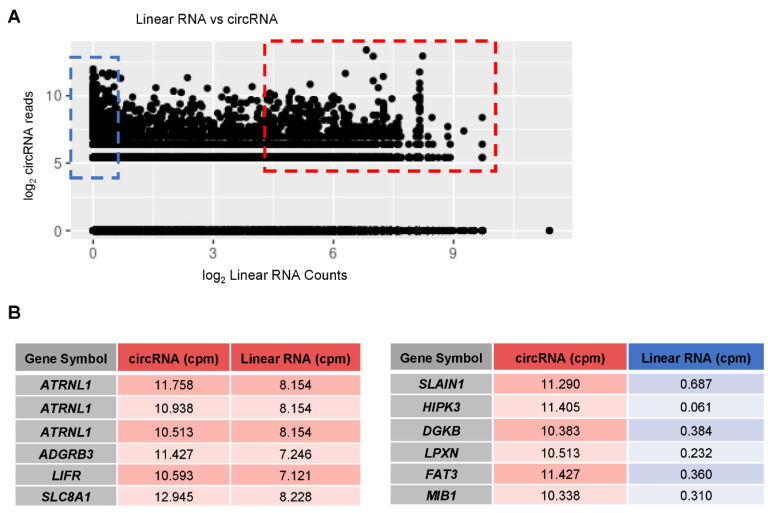
Linear vs. circRNA correlation plot. (**A**) Representative comparison of the number of circRNAs and linear reads from a single sample (GSE53697) to illustrate the lack of overall correlation seen between the expression of circRNAs and linear RNAs (r = 0.024). (**B**) Specific examples of parent genes that produce either high circRNA reads and high linear counts (inside the red box) or high circRNA reads and low linear counts (inside the blue box).

## Data Availability

The source data are publicly available from GSE53697, GSE110731, GSE67333, GSE104704, GSE161199, and PRJNA574438.

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
