# Peer review of "Systematic Identification of circRNAs in Alzheimer’s Disease"

_genes, 2021, doi:10.3390/genes12081258_

Round 1
Reviewer 1 Report
In this study, Cochran et al. identified circRNAs from existing Alzheimer’s and control RNAseq data sets across various brain regions and also plasma. Meta-analyses such as these are valuable for characterizing circRNAs in Alzheimer’s and are needed to help us better understand the role of circRNAs in both disease pathogenesis and under healthy conditions. Overall, the study & findings are clearly presented, but I have some major and minor comments below.
Major:
1) Supp Fig 1: It is surprising to see that those data sets with a lower average number of reads yielded higher numbers of circRNAs and those with a higher number of reads yielded lower numbers of circRNAs. This calls into question the sensitivity and precision of the method used. Have the authors considered an ensemble-based approach for circRNA detection? As CIRCexplorer2 uses a fragment-based approach for circRNA detection, using orthogonal approaches such as KNIFE would be helpful. If not an ensemble approach, an orthogonal post-hoc analysis of identified circRNAs, such as using ACValidator, may be another option to increase confidence in the identified circRNAs.
2) Was age, sex, PMI, etc, controlled for in each of the studies used?
3) It would be worth comparing results against any available data from RNase R treated samples that were subjected to RNAseq since that approach more effectively captures circRNAs.
4) Fig4C & Fig5A : are the differences across data sets (the two plasma data sets, as well as the plasma vs tissue data sets) attributable to the amount of RNAseq data generated for each?
5) It would be interesting to take a deeper look at circRNAs unique to each brain region since regions are differentially impacted in Alzheimer’s brains. Are any circRNAs unique to specific regions associated with known Alzheimer’s pathologies or pathogenesis? Pathway analysis of these uniquely expressed circRNAs would be helpful for assessing potential functional impacts.
6) Supp Table 1: which studies were each of the identified circRNAs identified in? Is the data shown an average of 2 studies if the respective circRNA was identified in 2 studies?
7) Line 205: what 3 circRNAs were shared across the 4 studies?
8) Line 248: what 2 circRNAs out of the identified 53 are related to AD?
9) Please include p-values in the 3 Supp tables. For Supp Tables 2 & 3, please indicate which brain region or tissue (plasma) that each circRNA was similarly identified in (along w/respective fold changes and p-values).
Minor:
1) Were plasma samples collected from live or ante-mortem subjects? (since this will influence interpretation of data)
2) Fig3: does this show all circRNAs with P<0.05?
Author Response
Comments and Suggestions for Authors
In this study, Cochran et al. identified circRNAs from existing Alzheimer’s and control RNAseq data sets across various brain regions and also plasma. Meta-analyses such as these are valuable for characterizing circRNAs in Alzheimer’s and are needed to help us better understand the role of circRNAs in both disease pathogenesis and under healthy conditions. Overall, the study & findings are clearly presented, but I have some major and minor comments below.
[AU]We appreciate the encouraging comments and advice from the Reviewer.
Major:
Supp Fig 1: It is surprising to see that those data sets with a lower average number of reads yielded higher numbers of circRNAs and those with a higher number of reads yielded lower numbers of circRNAs. This calls into question the sensitivity and precision of the method used. Have the authors considered an ensemble-based approach for circRNA detection? As CIRCexplorer2 uses a fragment-based approach for circRNA detection, using orthogonal approaches such as KNIFE would be helpful. If not an ensemble approach, an orthogonal post-hoc analysis of identified circRNAs, such as using ACValidator, may be another option to increase confidence in the identified circRNAs.
[AU] We thank the Reviewer for the question and suggestions. The believe what the differences in relative number of reads compared to numbers of circRNAs can be explained by the difference in brain relative to plasma samples, as in plasma there appears to be little circRNAs overall.
The suggestion that we use other methods to analyze circRNAs is well taken, as indeed there are many such software types available for circRNA analysis. We appreciate the suggestion of using ACValidator, which we had not considered previously, but appears to be helpful. We plan to implement it for future studies, as it would take several months’ time to reanalyze the current data using ACValidator. We did previously try to study circRNAs using KNIFE, but we found it cumbersome to use, and ‘de novo’ assembly did not appear possible using KNIFE. We have revised the manuscript text to suggest that other analytical approaches can yield valuable complementary data.
Was age, sex, PMI, etc, controlled for in each of the studies used?
[AU] We downloaded the data in bulk from GEO from these analyses, but the individual published studies did control for age, gender, and other parameters. In our own analysis we did not include these as covariates and instead relied on what the authors of the individual studies had done.
It would be worth comparing results against any available data from RNase R treated samples that were subjected to RNAseq since that approach more effectively captures circRNAs.
[AU] We fully agree with the Reviewer. Unfortunately, the biological samples used in this experiment were not available to us for molecular biology analysis. We mention this point in the revised discussion.
Fig4C & Fig5A : are the differences across data sets (the two plasma data sets, as well as the plasma vs tissue data sets) attributable to the amount of RNAseq data generated for each?
[AU] As the reviewer suggests, the size of the studies was very likely a factor in the differences across data sets, although it is difficult to know if such differences were solely attributable to study sizes. Given that studies of this type [RNA-seq without previous poly(A) selection] are not widely available, we only had these few studies for analysis. We agree that data sets that were more balanced in size would have been preferable. We also reflect this point in the revised text.
It would be interesting to take a deeper look at circRNAs unique to each brain region since regions are differentially impacted in Alzheimer’s brains. Are any circRNAs unique to specific regions associated with known Alzheimer’s pathologies or pathogenesis? Pathway analysis of these uniquely expressed circRNAs would be helpful for assessing potential functional impacts.
[AU] We agree with the Reviewer that the circRNAs differentially abundant in each brain region could be informative. Unfortunately, it is not possible at present to do meaningful pathway analysis, since we lack information on what these circRNAs do. Therefore, at the moment, assigning pathways to circRNAs based on what the host precursor RNA does might be misleading. As we learn more about the biological functions of circRNA, assigning pathways would be increasingly possible and informative. In the revised manuscript, we have specified more clearly the circRNAs that originate from each brain region, as specified in the source articles and in revised Table S1.
Supp Table 1: which studies were each of the identified circRNAs identified in? Is the data shown an average of 2 studies if the respective circRNA was identified in 2 studies?
[AU] The Reviewer is completely right, we did a poor job distinguishing the different studies across Supplemental Table S1. In the revised manuscript, we have subdivided Supplemental Table S1 into 6 spreadsheets, one per study. In the revised manuscript, we indicate these changes in our description of Table S1. The data shown are not an average of two studies, but simply every circRNA from every study. However, for Figure 7, if a circRNA was found significantly abundant in two studies, the greater relative difference is displayed.
Line 205: what 3 circRNAs were shared across the 4 studies?
[AU] We apologize for this oversight. The three circRNAs (now specified in the revised text) originate from gene loci ANKS1B, ARHGAP26, and DPYD.
Line 248: what 2 circRNAs out of the identified 53 are related to AD?
[AU] Again, we regret this omission and have added them to the revised text. The two circRNAs in question are derived from AD-associated gene loci KIF1B and DLG1.
Please include p-values in the 3 Supp tables. For Supp Tables 2 & 3, please indicate which brain region or tissue (plasma) that each circRNA was similarly identified in (along w/respective fold changes and p- values).
[AU] We appreciate this request. A p-values column has been added to the revised Supplemental Tables S2 and S3. To include the region of the brain where the given circRNAs were found was not possible within the time limit allowed, since we calculated the overlap based on Gene Symbol + Ensembl ID, so it was not possible to assign the gene segment from which each circRNA originated. We clarify in the revised text that these circRNAs were only present in at least one brain study and one plasma study.
Minor:
Were plasma samples collected from live or ante-mortem subjects? (since this will influence interpretation of data)
[AU] All the plasma samples were collected from live subjects. We have indicated this in the revised text.
Fig3: does this show all circRNAs with P<0.05?
[AU] Yes, Figure 3 shows all circRNAs with p<0.05.
Reviewer 2 Report
In this manuscript, Cochran et al. seek to identify circRNAs differently expressed in AD individuals that could be useful for future biomarker and functional studies. They use a robust bioinformatic approach to detect circRNA species from several RNAseq public datasets. This type of studies are important in order to identify circRNAs consistently altered among different cohorts, which would reinforce their contribution as well as help understand their link with the disease pathophysiology.
Below are some points that should be addressed by the authors:
- The distribution of data on Supplemental Table S1 is confusing. While increased and decreased circRNA are well separated in different columns, the list of circRNAs that correspond to each study is unclear, which makes it difficult to identify those species shared across studies.
- In section 3.3, the authors compare the overlap of circRNAs between plasma studies. The aggregate of dysregulated circRNAs (Supplemental Figure S2B) does not match the sum of those that appear in Figure 4C. Could you explain why is it due to? In the same section, there seems to be an inconsistency with the number of circRNAs: according to the tables, 79 circRNAs overlap in PRJNA and brain studies, however only 49 are depicted in the Venn’s diagram. It may be a typographical error, but please correct it.
- The correlation of dysregulated circRNAs between brains and plasma is indeed very interesting from the biomarkers point of view. I would suggest the authors to comment on the direction of the changes in each sample type (for the 5 mentioned circRNAs). I think it would also be worthy to explore if these circRNA changes have been validated in other cohorts, as it would support their significance for future biomarker and functional studies.
- The plot shown in Supplemental Figure 2A is the same as Figure 3A. Please update with the correct graph.
- In Figure 1A it would be helpful to add a column with the reference corresponding to each of the studies. In the same manner, I would encourage the authors to include in Figure 7 the appropriate references for the information provided in the last column (“Relation to Alzheimer’s disease”).
Author Response
Comments and Suggestions for Authors
In this manuscript, Cochran et al. seek to identify circRNAs differently expressed in AD individuals that could be useful for future biomarker and functional studies. They use a robust bioinformatic approach to detect circRNA species from several RNAseq public datasets. This type of studies are important in order to identify circRNAs consistently altered among different cohorts, which would reinforce their contribution as well as help understand their link with the disease pathophysiology.
[AU] We appreciate the Reviewer’s supportive comments.
The distribution of data on Supplemental Table S1 is confusing. While increased and decreased circRNA are well separated in different columns, the list of circRNAs that correspond to each study is unclear, which makes it difficult to identify those species shared across studies.
[AU] We could not agree more with the Reviewer. We now display Table S1 as a set of 6 independent spreadsheets, one per study.
In section 3.3, the authors compare the overlap of circRNAs between plasma studies. The aggregate of dysregulated circRNAs (Supplemental Figure S2B) does not match the sum of those that appear in Figure 4C. Could you explain why is it due to? In the same section, there seems to be an inconsistency with the number of circRNAs: according to the tables, 79 circRNAs overlap in PRJNA and brain studies, however only 49 are depicted in the Venn’s diagram. It may be a typographical error, but please correct it.
[AU] We appreciate the Reviewer’s careful read. The discrepancies are due to the fact that ‘dysregulation’ encompasses both ‘upregulation’ and ‘downregulation’. That is to say, a circRNA may be found dysregulated in 2 studies, but be upregulated in one study and down regulated in the other. As a result, the numbers change slightly depending on the comparison type. The Venn diagrams in Figure 4C have a different overall sum because the conditions of overlap are more stringent than in Supplemental Figure S2B, as we required them to be dysregulated in the same direction, either upregulated in both studies, or downregulated in both studies. We have clarified these points in the revised text.
The correlation of dysregulated circRNAs between brains and plasma is indeed very interesting from the biomarkers point of view. I would suggest the authors to comment on the direction of the changes in each sample type (for the 5 mentioned circRNAs). I think it would also be worthy to explore if these circRNA changes have been validated in other cohorts, as it would support their significance for future biomarker and functional studies.
[AU] We totally agree with the Reviewer. It would be worth exploring if these circRNA changes are seen in other cohorts. No other cohorts of this type were available to us at this moment, as very few studies have sequenced RNA without previously selecting for poly(A) RNA (this step of course, eliminates circRNAs from the analysis). As our work progresses, we will examine the biomarker value of these circRNAs. The direction of changes for the circRNAs in Figure 7 had been included in the original version (5th column), but was not very apparent. Following the reviewer’s advice, we have added text to be sure the reader is aware of this information.
The plot shown in Supplemental Figure 2A is the same as Figure 3A. Please update with the correct graph.
[AU] We appreciate the Reviewer’s catching this mistake. We have replaced Figure 3A with the correct graph.
In Figure 1A it would be helpful to add a column with the reference corresponding to each of the studies. In the same manner, I would encourage the authors to include in Figure 7 the appropriate references for the information provided in the last column (“Relation to Alzheimer’s disease”).
[AU] Yes, we absolutely agree. We have added the requested column in Table 1 Add to legend. For Figure 7, the relevant references have been added in a separate ‘Refs.’ column.
Round 2
Reviewer 2 Report
I would like to thank the authors for taking into account my suggestions and addressing my concerns. The additional information provided and distribution of the tables is very helpful to better understand the results of the study.